# CNAS: Channel-Level Neural Architecture Search

## Abstract

There is growing interest in automating designing good neural network architectures. The NAS methods proposed recently have significantly reduced architecture search cost by sharing parameters, but there is still a challenging problem of designing search space. We consider search space is typically defined with its shape and a set of operations and propose a channel-level architecture search (CNAS) method using only a fixed type of operation. The resulting architecture is sparse in terms of channel and has different topology at different cell. The experimental results for CIFAR-10 and ImageNet show that a fine-granular and sparse model searched by CNAS achieves very competitive performance with dense models searched by the existing methods.

## 1 Introduction

Nowadays, deep neural networks (DNNs) are used extensively and successfully in many fields and applications such as computer vision, speech recognition, machine translation, and automated vehicles. Designing DNNs often requires significant architecture engineering, a large amount of trial and error by experts. Although transfer learning is widely used to save the efforts required for designing good architectures of DNNs from scratch, it is not always possible to use.

Recently, there is growing interest in automating designing good neural network architectures (30; 31; 20; 24; 21; 15; 3; 28; 14; 2; 22; 7; 27; 4; 29; 10). Most of them can be categorized into reinforcement learning-based (RL) methods, evolutionary algorithm-based (EV) methods, hypernetwork-based (HY) methods, and gradient-based (GR) methods, in terms of the search algorithm.

RL methods (30; 31; 20; 24) use a controller model that enumerates a bunch of candidate models, which are trained for a fixed number of epochs from scratch, and then, is updated using the validation accuracies of the candidate models evaluated on a validation set. To reduce the search space of candidate models, some of them (31; 20) assume each model is composed of multiple convolutional layers called *cells* having the same architecture and focuses on searching for the best cell architecture. For example, in NASNet (31), a cell is composed of five blocks, and each block composed of two operations, which are selected among a set of various convolution and pooling operations by the controller model. To reduce the search space, NASNet also transfers the learned architecture for a small dataset (e.g., CIFAR-10) to a large dataset (e.g., ImageNet). To optimize an architecture with less amount of computation, ENAS (20) exploits parameter (weight) sharing, which avoids training each candidate model from scratch by sharing the weights of candidate models. It constructs a large computational graph, where each subgraph represents the architecture of a candidate model, and the controller model is trained to search for a subgraph corresponding to a good candidate model.

EV methods (23; 1; 12; 17; 26; 21; 15) also have been extensively studied. AmoebaNet (21) uses the same search space with NASNet, but searches a good cell architecture based on evolutionary algorithm instead of RL controller. The population is initialized with models with random architectures, and some models are sampled from the population. The model with the highest validation fitness within the samples is selected as the parent (i.e., exploitation), and a child having a mutation in terms of operations and skip connections is constructed from the parent (i.e., exploration). Hierarchical NAS (15) uses hierarchical representation for cell architecture where smaller graph motifs are used as building blocks to form larger motifs, instead of flat representation. Unfortunately, most of RL and EV methods, except ENAS, require an enormous amount of computing power for training thousands

of child models. They usually need hundreds or thousands of GPU days for architecture search, which is almost impossible for a typical machine learning practitioner.

HY methods and GR methods avoid such a large cost of architecture search by sharing parameters as in ENAS (20). HY methods (3; 28) bypass fully training candidate models by instead training an auxiliary model, a HyperNet (8), to dynamically and directly generate the weights of a candidate model. SMASH (3) generates an architecture of an entire network (i.e., macro search) in terms of the hyperparameters of filters (e.g., number, size) with fixing the type of operation in the HyperNet space, while GHN (28) generates an architecture of a cell (i.e., micro search) in terms of operations in the NAS search space.

GR methods (22; 7; 27; 4; 29; 10) do not rely on controllers, evolutionary algorithm, and hypernetworks, but exploit gradient descent on network architectures, which can significantly improve the speed of NAS. They basically relax the search space to be continuous, so that the architecture can be optimized with respect to its validation set performance by gradient descent. Here, search space corresponds to a parent network, and a child network (subgraph) can be derived from the parent network by gradient descent. Most of GR methods focus on searching a good cell architecture in terms of operations and repeating the same architecture as in NASNet. After architecture search, they should usually re-train the candidate architecture snapshot from scratch using the training set due to inconsistency between the performance of derived child networks and converged parent networks.

As described above, one of the major trends in NAS is exploiting the concept of parameter sharing through hypernetwork or gradient descent in order to reduce the cost (i.e., GPU days) of NAS. By parameter sharing, HY and GR methods can automatically optimize an architecture that can achieve the state-of-the-art performance on CIFAR-10 and ImageNet just within a few days (as summarized in Table 4). However, there is still a challenging problem in the above architecture search methods: designing search space. In principle, the search space should be large and expressive enough to capture a diverse set of promising candidate models, and at the same time, should be small enough to train with the limited amount of resources and time (2). Some methods (24; 15; 22) addressed that defining search space is extremely important for the performance of neural architecture search. The problem about designing search space may not be solved at once. The search space of the existing NAS methods is typically defined with a shape of the overall network and a set of operations such as identity, normal convolution, separable convolution, average pooling, and max pooling. Many of them follow the NASNet search space for the shape of the network (i.e., stacking cells) and define their own set of operations. Since the number of possible types of operations for search space is limited due to the search cost, the set of operations used itself may have a large impact on the performance of architecture search.

In this paper, we investigate the possibility of achieving competitive performance with the state-of-the-art architecture search methods with using a fixed type of operation. To achieve such a performance, we focus on the *sparsity* of a model. A candidate model in the existing methods has multiple types of operations connected with each other via skip connections, and each operation takes the entire feature maps (called channel) of certain previous nodes or cells as input and returns its entire resulting channels as output. Thus, the candidate model can be regarded as a dense model in term of input and output channels of the operations. We propose a channel-level neural architecture search (CNAS) method that regards channels as vertices and a single fixed operation as edges and searches for a good architecture by gradient descent. The resulting model is sparse in terms of channels. CNAS uses the existing shape of search space (e.g., NASNet), but performs macro search. Thus, the resulting architecture has different topology at different cells. In CNAS, the final sparse architecture can be searched quickly due to its simplicity, and at the same time, can compensate for the disadvantage of using homogeneous operation due to its sparsity. For CIFAR-10, CNAS searches for the architecture in 1.1 GPU days, which achieves 2.28% test error with 4.6 million parameters and autoaugment.

The rest of the paper is organized as follows. Section 2 explains our method CNAS. Section 3 shows the experimental results, and Section 4 summarizes the characteristics of related methods. Section 5 concludes this paper.

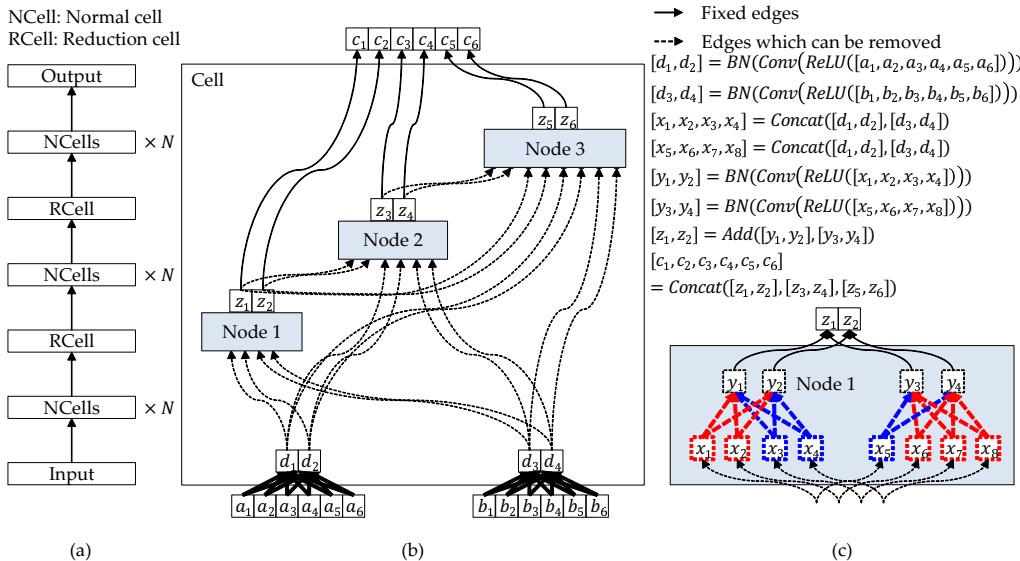

Figure 1: Diagram of search space of CNAS. (a) Shape of search space. We usually set $N = 6$. (b) Structure of an example cell of the model. The small white squares indicate channels. Both $\{a_i\}$ and $\{b_i\}$ are the input channels of the cell. $\{c_i\}$ are the output channels of the cell. The solid edges are fixed operations, and the dotted edges can be removed depending on the existence of the input channels $\{x_i\}$. The thick edges indicate a series of operations, batch normalization (BN), depthwise separable 3x3 convolution (Conv), and Relu, while the thin edges simple operations such as concatenation and add. (c) Structure of an example node in the cell. Each node has two operations (of the same type). $\{x_i\}$ are input channels of the node, while $\{y_i\}$ are intermediate output channels. The red input channels and red thick dotted edges are removed, while the blue input channels and blue thick dotted edges remain as a part of the final sparse model.

## 2 CNAS METHOD

Since we focus on investigating the possibility of architecture search relying on sparsity instead of the combination of operations in this paper, we mainly use the structure of NASNet for the shape of search space, which is composed of normal cells and reduction cells, and each cell is again composed of submodules called *nodes* (blocks in NASNet). In Section 3, we will show the result of CNAS using different shape of search space, in particular, the structure of DenseNet. Figure 1 shows the diagram of the search space of CNAS.

### 2.1 SEARCH SPACE

The CNAS method consists of the following three steps: (1) Train the one-shot (i.e., full-edges) model in a fixed number of epochs to make it predictive of the validation accuracies of sparse models. (2) Search the most promising sparse model satisfying a given sparsity based on a criteria (e.g., Taylor) by zeroing out less important channels. (3) Re-train the most promising model from scratch (called CNAS-R) or fine-tune it (called CNAS-W) and then evaluate the final model on the test dataset.

In CNAS, a vertex in a cell or a node is a single channel, and an edge is an operation. In Figure 1(b) and (c), the thick edges are non-trivial operations involving convolution, where the solid thick ones are for pre-processing as in other methods, and the dotted solid thick ones are the part that can be changed by architecture search. The type of operation used in CNAS is fixed as a specific one, depthwise separable 3x3 convolution since operations are not the target of architecture search. In contrast, the number of types of operations in the existing NAS methods is at least several, and the types of operations are designed differently depending on the method.

In Figure 1(c), zeroing out less important channels of red squares also removes their outgoing edges to the next layer $\{y_i\}$, i.e., does not apply the operation $BN(Conv(ReLU(\cdot)))$ to the red squares. In terms of low-level implementation, CNAS performs partial matrix operations between $\{y_1, y_2\}$ and $\{x_3, x_4\}$ and between $\{y_3, y_4\}$ and $\{x_5\}$. Since the red edges do not need to be calculated, the corresponding convolution kernels are also not necessary. Thus, the number of weight parameters between $\{x_i\}$ and $\{y_i\}$ is reduced by $\frac{3}{8}$ in Node 1. That is, channel-level architecture search makes the model sparse. If $\{x_1, x_2, x_3, x_4\}$ all are removed, then $\{y_1, y_2\}$ are also removed, and only $\{y_3, y_4\}$ is added to $\{z_1, z_2\}$. In general, the input channels of nodes (i.e., dotted squares) are removed differently depending on whether the cell is close to input data or close to the output layer. Thus, after architecture search, each cell in CNAS has different architecture in terms of the topology of vertices and edges.

## 2.2 SEARCHING THE MOST PROMISING SPARSE MODEL

We search for the most promising sparse model satisfying a given sparsity $\rho$ by zeroing out less important channels. Here, $\rho$ indicates $\frac{|W^*|}{|W|}$, where $|W|$ is the number of weight parameters of the one-shot model, and $|W^*|$ that of the final sparse model ($0.0 \le \rho \le 1.0$). As the criteria for evaluating the importance of channels, we adopt Taylor expansion (18). Algorithm 1 shows the outline of the evaluation. We denote the vector of entire input channels $\{x_i\}$ in the one-shot model as $X$ and the length of $X$ as $|X|$. Likewise, we denote the vector of entire gradients of $X$ after a single minibatch as $\Delta X = \{\delta x_i\}$. When calculating gradients, we use the current sparse model $W'$, which is initially the same with $W$, and the parameters of $W'$ are not updated. $X'$ and $\Delta X'$ are the vector of entire input channels and their gradients in the current sparse model, respectively. The saliency vector $S$ has the same length with $|X'|$ and is initialized with zeros. We consider $m$ minibatches for the input dataset $D$. Then, we get $X'$ and $\Delta X'$ in each minibatch and calculate Taylor expansion using element-wise multiplication between both. The smaller $x_i \odot \delta x_i$ is, the larger the value $\frac{1}{x_i \odot \delta x_i}$ is. The dimension of the value is reduced to a single value, which is again accumulated to the corresponding saliency value in $S$. Then, we normalize $S$ by applying layer-wise L2-normalization.

---

**Algorithm 1:** Calculation of Taylor expansion for CNAS

1 **for each** $D_k \in [D_1, \cdots, D_m]$ **do**
2 $\quad$ $X', \Delta X' \leftarrow$ **ForwardAndBackpropagation**$(D_k, W')$
3 $\quad$ $S \leftarrow S +$ **DimensionReduction**$(\frac{1}{X' \odot \Delta X'}, |X'|)$
4 $S \leftarrow$ **Normalization**$(S)$

---

After calculating the saliency vector $S$, we gradually zero out the input channels $\{x_i\}$ having the largest values, i.e., least important channels, among the remaining input channels. We let the ratio of zeroing out $\gamma$ ($0 < \gamma < 1$). We typically use $\gamma = 0.1$, which means removing 10% input channels of the remaining input channels at each iteration. Thus, the number of input channels becomes $0.9|X|$ after the first iteration and $0.81|X|$ after the second iteration. We perform fine-tuning of a single epoch for the current sparse model $W'$ between iterations. As the iteration goes on, the model becomes sparser and sparser. We stop the iterations when the sparsity of $W'$ reaches the given $\rho$. After finding the final sparse model $W^*$, we can initialize the parameters of $W^*$ and re-train the model (called CNAS-R), or just fine-tune the parameters of $W^*$ (called CNAS-W).

We incorporate spatial dropout (25) at training the one-shot model or the final sparse model in order to make that the model more robust. We do not use path dropout used in ENAS (20) and One-Shot (2) since it is too coarse to incorporate for our channel-level search. We also do not use conventional drouout (9) since it is too fine-grained to apply. Although One-Shot (2) consider the co-adaptation issue in which zeroing out operations from the one-shot model can cause the quality of the model's prediction to degrade severely, we do not need to consider it since the final sparse model is obtained through gradually zeroing out by the ratio $\gamma$.

We check the correlation between the one-shot model and the final sparse model in terms of performance (test error). We generate 27 pairs of one-shot models of three cells and five nodes per cell with different initialization and train them for 150 epochs. Then, we search a single final sparse model from each one-shot model and train them for 310 epochs. Figure 2(a) shows a strong correlation

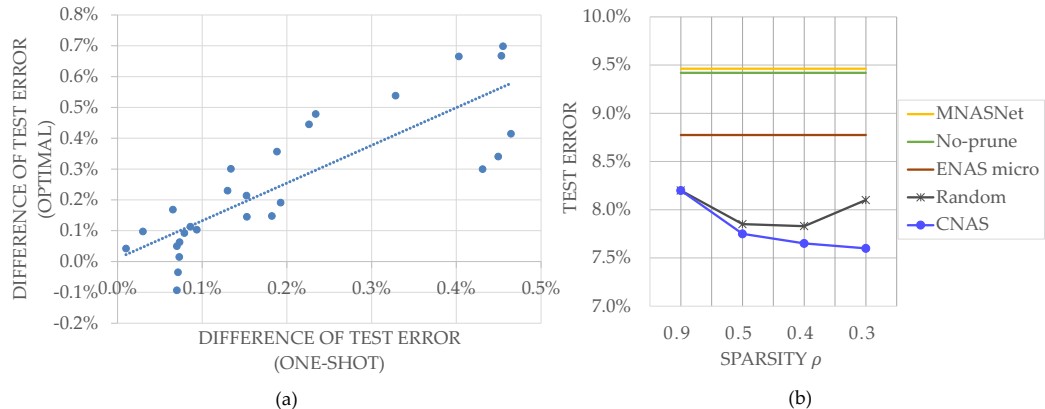

Figure 2: Results of sparse models. (a) Correlation between the one-shot and the final sparse model in terms of test error. (b) Comparison among no-pruning, randomly pruning, and our pruning (all have the same number of parameters of 0.15 M).

between the one-shot and the final sparse model in terms of test error. We let $E(\cdot)$ a test error. X-axis means $E(W_1) - E(W_2)$ where $W_1$ and $W_2$ are a pair of one-shot models s.t. $E(W_1) > E(W_2)$. Y-axis means $E(W_1^*) - E(W_2^*)$ where $W_1^*$ and $W_2^*$ are the final sparse models of $W_1$ and $W_2$, respectively. There are 27 points in the figure, and only two points are located below 0.0 at Y-axis. For the remaining 25 points, if $W_1$ is better than $W_2$, then $W_1^*$ is also better than $W_2^*$. The test error of the one-shot model is computed before architecture search. There is no fine-tuning for the one-shot model. The test error of the optimal model is computed after fine-tuning. The correlation coefficient between X-axis and Y-axis is about 0.83. It means the way of searching the final sparse model is stable.

## 2.3 TOPOLOGICAL PROPERTIES OF THE FINAL SPARSE MODEL

We describe the topological properties of the final sparse model $W^*$ after channel-level architecture search. Table 1 shows the statistics of $W^*$ compared with those of the one-shot model for CIFAR-10. Due to the space limit, we show only the top three cells, the bottom three cells and two reduction cells among 20 cells. From the statistics, we address two properties. First, the input channels in reduction cells are not removed as much as in other top and bottom cells. For example, in Cell 18, $|X^*|$ is smaller than one-fifth of $|X|$. In contrast, in Cell 14, $|X^*| = 5,233$ is almost the same with $|X| = 5,760$ In reduction cells, the height and width of a channel is reduced by half, while the number of channels is increased by two times. As a result, the amount of information is reduced by half. Keeping input channels at reduction cells seems to be due to compensating the loss of information to achieve the lower error. Second, $|z \rightarrow x|$ is extremely low compared with $|d \rightarrow x|$ in the top cells, whereas $|z \rightarrow x|$ is similar with $|d \rightarrow x|$ in the bottom cells. The former means that there is almost no edge among the nodes in the top cells and so the nodes are located horizontally, each of which is doing its own task. The latter means that there are a lot of edges among the nodes in the bottom cells and so the nodes are located vertically and horizontally as in Figure 1 with the necessity of aggressive abstraction.

## 3 EXPERIMENTS

We use CIFAR-10 (13) and ImageNet (6) for our experiments. For training the one-shot model of CIFAR-10, we use 150 epochs with the Nesterov momentum (19) 0.9. We used a cosine learning rate schedule (16) with the initial learning rate $l_{max} = 0.05$, the minimum learning rate $l_{min} = 0.0001$, the initial number of epochs $T_0 = 10$, and the multiplication factor $T_{mul} = 2$ and $\ell_2$ weight decay of $2 \times 10^{-4}$. We train the final sparse model by using the same setting with the one-shot model, except the number of epochs, which is set to 630. For ImageNet, we use the same final sparse model for CIFAR-10 only after adding two more stem convolution layers and modifying the fully connected layer to handle the different number of outputs. We use 250 epochs for training the modified sparse

| one-shot model | | | | final sparse model ($\rho = 0.46$) | | | |
|---|---|---|---|---|---|---|---|
| cell | $|x|$ | $|y|$ | # params | $|x^*|$ | # params | $|d \rightarrow x|$ | $|z \rightarrow x|$ |
| 20 | 5,760 | 1,440 | 1,088,640 | 795 | 328,995 | 791 | 4 |
| 19 | 5,760 | 1,440 | 1,088,640 | 646 | 306,198 | 644 | 2 |
| 18 | 5,760 | 1,440 | 1,088,640 | 1,010 | 361,890 | 969 | 41 |
| 14 | 5,760 | 1,440 | 984,960 | 5,233 | 904,329 | 2,537 | 2,696 |
| 7 | 2,880 | 720 | 259,200 | 2,244 | 207,684 | 1,258 | 986 |
| 3 | 1,440 | 360 | 77,760 | 284 | 25,740 | 170 | 114 |
| 2 | 1,440 | 360 | 75,168 | 653 | 39,753 | 367 | 286 |
| 1 | 1,440 | 360 | 72,576 | 905 | 48,501 | 531 | 374 |
| Total | 69,120 | 17,280 | 9,880,704 | 26,038 | 4,548,078 | 17,322 | 8,716 |

Table 1: Statistics of the one-shot model and the final sparse model in CNAS for CIFAR-10. The number of cells is 20, and the number of nodes per cell is five. $|X|$ and $|Y|$ are the numbers of input channels and output channels in all nodes of the one-shot model, respectively. $|X^*|$ is the number of input channels of the final sparse model. $z$ means the output channels of nodes, and $d$ the input channels of cells after being preprocessed as in Figure 1(b). Cell 7 and cell 14 are reduction cells, and other cells are all normal cells. $|d \rightarrow x|$ and $|z \rightarrow x|$ are the numbers of the edges from $\{d_i\}$ to $\{x_j\}$ and from $\{z_i\}$ to $\{x_j\}$ in Figure 1, respectively.

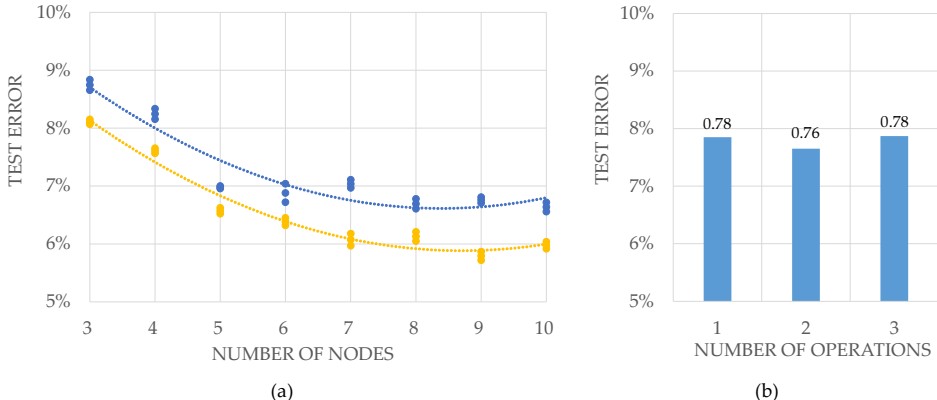

(a)  (b)

Figure 3: Test error of CNAS with different settings. (a) Varying the number of nodes per cell. The number of parameters of all models is 1.1 M regardless of the number of nodes. (b) Varying the number of operations per node. The number of parameters of all models is 0.15 M regardless of the number of operations.

model with the Nesterov momentum 0.9 and the learning rate 0.05, which is decayed by a factor 0.98 after each epoch. All test errors in the result are the mean values of three evaluations.

## 3.1 EVALUATION OF CNAS VARYING THE NUMBER OF NODES AND OPERATIONS

We check the performance of CNAS models having the same number of parameters while varying the number of nodes per cell or varying the number of operations per node. We use the one-shot model of two normal cells and one reduction cell. Figure 3(a) shows the result of CNAS models, which all have 1.1 M parameters, but different number of nodes per cell. The test error tends to be decreased as the number of nodes per cell increases (i.e., model becomes sparser), but slightly increases when the number of nodes is ten (i.e., too sparse). Figure 3(b) shows the result of CNAS models, which all have 0.15 M parameters, but different number of operations (of the same type) per cell. Although we use two operations per cell to follow the convention of NASNet, the difference in test error among three settings is quite small. This is mainly due to using a single type of operation (edge).

### 3.2 COMPARISON AMONG DIFFERENT PRUNING METHODS FOR CNAS

In this section, we evaluate the performance of no-pruning, randomly pruning, and our pruning (in Section 2.2). Here, no-pruning means making a small one-shot model of 0.15 M parameters and training the model without the pruning step. Randomly pruning means pruning each cell randomly with a given sparsity (e.g., 0.9, 0.5, 0.4, 0.3) and making the final model all have the number of parameters of 0.15 M. Thus, this setting does not have different topology at each cell, but rather has similar topology. Our pruning also has the 0.15 M parameters, but different topology at each cell as in Table 1. We use CIFAR-10 for comparison. Figure 2(b) shows the results of three settings. Among them, no-pruning shows the worst performance, while our pruning of CNAS shows the best performance. It means that a sparse model in terms of channel improves the performance compared with a dense model, and at the same time, different topology at each cell is important to achieve the better performance. Just for reference, we add the results of MNASNet (24) and ENAS (20) having the same number of parameters of 0.15 M. Both No-prune and ENAS are dense models in terms of channel, but ENAS shows a better performance than No-prune due to its various operations in search space. MNASNet (24) shows slightly worse performance than No-prune since its architecture is the one optimized for ImageNet.

### 3.3 COMPARISON WITH OTHER METHODS

In this section, we present the comparison result with the state-of-the-art methods for CIFAR-10 and ImageNet. Table 2 shows the model size, test errors and GPU days (for architecture search methods) for CIFAR-10. For CNAS, we measure the first two steps, i.e., training the one-shot model and searching the final sparse model, as the GPU days for architecture search. The CNAS model used in the comparison is the same as the final sparse model in Table 1. Overall, CNAS achieves very competitive performance with only 1.1 GPU days and a moderate number of parameters (4.6 M) among all the methods compared. CNAS with autoaugment (5) can further improve the performance up to 2.28% test error.

Table 3 shows the comparison result for ImageNet. The model size of CNAS slightly increases to 5.7 M due to adding two stem convolution layers to and modifying the fully connected layer of the final sparse model in Table 1. We denote this model just as CNAS. Overall, CNAS achieves comparable performance with other methods but does not show very competitive performance as in CIFAR-10. It is probably because we use the same final sparse model obtained from CIFAR-10 for ImageNet due to the limit of evaluation time. Searching and training an inherent final sparse model from ImageNet may further improve the performance with spending more GPU days for architecture search.

### 3.4 USING DENSENET-LIKE SEARCH SPACE FOR CNAS

In this section, we apply CNAS to a different shape of search space. In particular, we use DenseNet-BC (11) search space instead of NASNet search space. Figure 4 shows the diagram of the search space for CNAS. In Figure 4(a), each dense block consists of the 19 bottleneck layers, and there are transition layers between dense blocks. In each bottleneck layer in Figure 4(b), zeroing out less important channels in red squares also removes their outgoing edges to the next layer $\{y_i\}$. $x$ are concatenated to $z$ in output to make skip connection. The number of $x$ increases as the bottleneck layer number increases as in DenseNet. In Table 2, CNAS-R (DenseNet-BC) outperforms the original DenseNet-BC with the same number of parameters of 0.8 M. This means our CNAS method is effective in not only NASNet search space, but also different shapes of search space.

We note that the performance of CNAS-R (DenseNet-BC) with 4.6 M parameters is worse than that of CNAS-R using NASNet search space in Table 2. This means the shape of the search space of NASNet itself is superior to that of DenseNet.

## 4 RELATED WORK

We have briefly explained the recently proposed NAS methods according to the search algorithm in Section 1. Table 4 summarizes their characteristics in terms of not only search algorithm, but also search space, search range and how to generate candidate parameters.

| Methods | # params ($\times 10^6$) | GPU days for architecture search | Test error (%) |
|---|---|---|---|
| SMASH (3) | 16.0 | 1.5 | 4.03 |
| NAS (30) | 37.4 | 16,800 | 3.65 |
| Hierarchical NAS (15) | 61.3 | 300 | 3.63 |
| Progressive NAS (14) | 3.2 | 150 | 3.63 |
| One-shot Top (F=64) (2) | 10.4±1.0 | 3.3+$\alpha$ | 4.1±0.2 |
| One-shot Small (F=64) (2) | 5.0±0.2 | 3.3+$\alpha$ | 4.0±0.1 |
| NASNet-A (31) | 3.3 | 1,350 | 3.41 |
| ENAS macro (20) | 38.0 | 0.45 | 3.87 |
| ENAS micro (20) | 4.6 | 0.45 | 3.54 |
| GDAS (7) | 3.4 | 0.21 | 3.87 |
| GDAS (FRC) (7) | 2.5 | 0.17 | 3.75 |
| DARTS + cutout (22) | 2.9 | 1.5 | 2.94 |
| DARTS + cutout (22) | 3.4 | 4 | 2.83±0.06 |
| AmoebaNet-A + cutout (21) | 3.2 | 3150 | 3.3±0.06 |
| AmoebaNet-B + cutout (21) | 2.8 | 3150 | 2.55±0.06 |
| Petridish cell + cutout (10) | 3.2 | 5 | 2.75 |
| SNAS (mild constraint) + cutout (27) | 2.9 | 1.5 | 2.98 |
| GHN Top-Best + cutout (28) | 5.7 | 0.84 | 2.84±0.07 |
| DSO-NAS-share + cutout (29) | 3.0 | 1 | 2.84±0.07 |
| DenseNet-BC (11) | 0.8 | - | 4.51 |
| CNAS-R (DenseNet-BC) | 0.8 | 0.18 | 4.28 |
| CNAS-R (DenseNet-BC) | 4.6 | 0.67 | 3.97 |
| CNAS-R | 4.6 | 1.1 | 3.49 |
| CNAS-W | 4.6 | 1.1 | 3.40 |
| CNAS-R + cutout | 4.6 | 1.1 | 2.94 |
| CNAS-W + cutout | 4.6 | 1.1 | 2.77 |
| CNAS-R + autoaugmented (5) | 4.6 | 1.1 | 2.39 |
| CNAS-W + autoaugmented (5) | 4.6 | 1.1 | 2.28 |

Table 2: Comparison results among the state-of-the-art architecture search methods for CIFAR-10.

| Methods | # params ($\times 10^6$) | GPU days for architecture search | Top-1 | Top-5 |
|---|---|---|---|---|
| NASNet-A (31) | 5.3 | 1,350 | 74 | 91.3 |
| DARTS (22) | 4.9 | 4 | 73.1 | 91 |
| MNASNet-92 (24) | 4.4 | 1666 | 74.8 | 92.1 |
| One-shot Top (F=24) (2) | 6.8±0.9 | 3.3+$\alpha$ | 73.8±0.4 | - |
| One-shot small (F=32) (2) | 5.1±1.5 | 3.3+$\alpha$ | 74.2±0.3 | - |
| AmoebaNet-A (21) | 5.1 | 3150 | 74.5 | 92 |
| AmoebaNet-C (21) | 6.4 | 3150 | 75.7 | 92.4 |
| GDAS (7) | 5.3 | 0.21 | 74 | 91.5 |
| GDAS (FRC) (7) | 4.4 | 0.17 | 72.5 | 90.9 |
| Petridish cell (10) | 4.8 | 5 | 73.7 | - |
| SNAS + mild constraint (27) | 4.3 | 1.5 | 72.7 | 90.8 |
| GHN Top-Best (28) | 6.1 | 0.84 | 73 | 91.3 |
| DSO-NAS (29) | 4.7 | 1 | 73.8 | 91.4 |
| DSO-NAS-share (29) | 4.8 | 6 | 74.6 | 91.6 |
| CNAS-R | 5.7 | 1.1 | 74 | 91.9 |

Table 3: Comparison results among the state-of-the-art architecture search methods for ImageNet.

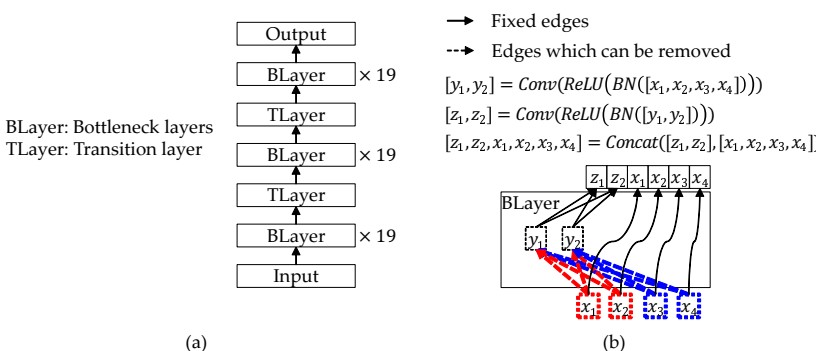

(a)          (b)

Figure 4: Diagram of DenseNet-like search space for CNAS. The meaning of boxes and edges is the same in Figure 1.

| Model | Search space | Search algorithm | Search range | Candidate parameters | Search cost (GPU days) |
|---|---|---|---|---|---|
| NAS (30) | OS | RL | Macro | Independent | 16800 |
| NASNet (31) | OS | RL | Micro | Independent | 1350 |
| ENAS (20) | OS | RL | Micro,Macro | Shared | 0.45,0.32 |
| MNASNet (24) | OS | RL | Macro | Independent | 1666* |
| AmoebaNet (21) | OS | EV | Micro | Independent | 3150 |
| Hierarchical NAS (15) | OS | EV | Micro | Independent | 300 |
| SMASH (3) | HS | HY | Macro | Dynamic | 1.5 |
| GHN (28) | OS | HY | Micro | Dynamic | 0.84 |
| Progressive NAS (14) | OS | PD | Micro | Independent | 225 |
| One-shot (2) | OS | RS | Micro | Shared | 3.3+a |
| DARTS (22) | OS | GR | Micro | Shared | 1.5 |
| GDAS (7) | OS | GR | Micro | Shared | 0.17 |
| SNAS (27) | OS | GR | Micro | Shared | 1.5 |
| ProxylessNAS (4) | OS | GR,RL | Macro | Shared | 8.3* |
| DSO-NAS (29) | OS | GR | Micro,Macro | Shared | 1,6* |
| Petridish (10) | OS | GR | Micro,Macro | Shared | 5,5 |
| CNAS | CS | GR | Macro | Shared | 1.1 |

Table 4: Characteristics of recently proposed NAS methods. Search space: OS (Operations and skip connections), HS (Hyperparameters of filters and skip connections) and CS (Channels and skip connections). Search algorithm: PD (Performance prediction) and RS (Random Search). Search range: Micro (Cell-level search and repeating the structure), Macro (Network-level search), Both (Cell- and Network-level search). Candidate parameters: Independent (generating parameters Independently), Shared (exploiting Shared parameters) and Dynamic (generating parameters dynamically). Search cost (GPU-days): All numbers except (*) are the costs of searching for CIFAR-10. (*) mean the costs for ImageNet due to no numbers for CIFAR-10 (i.e., no proxy) in the corresponding papers.

## 5 CONCLUSIONS

In this paper, we proposed a channel-level neural architecture search (CNAS) method that considers channels instead of operations for search space of NAS. It only uses a single fixed type of operation and instead focuses on searching for a good sparse architecture in terms of channel. The resulting sparse model has different topology at different cell. In particular, the nodes in the bottom cells are located vertically and horizontally with the necessity of aggressive abstraction, but the ones in the top cells are located horizontally for doing their own tasks. For CIFAR-10, CNAS achieves 2.28% test error with 4.6 million parameters using the architecture searched for only 1.1 GPU days. We also showed that CNAS is effective in not only NASNet search space, but also different shapes of search space.

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
