# OpenReview forum: "CNAS: Channel-Level Neural Architecture Search"
_ICLR.cc/2020/Conference — Reject_

### Official Review · AnonReviewer2 · 2019-10-22
**Official Blind Review #2**

**Rating:** 3

**Review:**

This paper proposes a channel pruning approach based one-shot neural architecture search (NAS). Unlike other NAS works that mostly search for operations/connections and topologies, this paper focuses on pruning channels for a fixed network.

In general, the idea of channel pruning has been extensively studied in previous works, and the channel pruning search algorithm is very similar previous one-shot NAS framework. The results on CIFAR-10 are reasonably good, but the results on ImageNet are not competitive to other NAS works.

Here are some more comments:

1. This paper is more like a new automated pruning technique rather than a new NAS technique. Therefore, I recommend the authors compare this technique with other pruning techniques, such as NetAdapt (https://arxiv.org/abs/1804.03230 ) and AMC (https://arxiv.org/abs/1802.03494).

2. The baseline model described in Figure 1 is quite limited. It would be helpful if the authors can also apply this pruning technique to other types of models (such as NASNet-A/MNASNet-92 from your Table 3,  or mobilenets used in NetAdapt/AMC papers).

3. Section 2.2 and Algorithm 1 is difficult to follow. It is not clear how Taylor expansion is carried out, and how saliency vector S is used. I recommend the authors expanding Algorithm 1 to include more details.

4. Figure 2(b) shows random pruning leads to better results than no-pruning. This is kind of counter-intuitive, could you give more details about your settings and explanation?

5. There are some minor errors: (1) Figure 1 [y1, y2] should be [z1, z2], and [y3, y4] should be [z3, z4];  (2) At the end of section 2.1, the number of weights in node 1 should be reduced by 4/8 instead of 3/8.


**Experience Assessment:**

I have published one or two papers in this area.

**Review Assessment: Checking Correctness Of Derivations And Theory:**

N/A

**Review Assessment: Checking Correctness Of Experiments:**

I assessed the sensibility of the experiments.

**Review Assessment: Thoroughness In Paper Reading:**

I read the paper at least twice and used my best judgement in assessing the paper.

---

### Official Review · AnonReviewer1 · 2019-10-23
**Official Blind Review #1**

**Rating:** 1

**Review:**

The paper models the neural architecture search (NAS) as a network pruning problem, and propose a method to sparsify the super-net during the search of architectures.

Overall, the novelty in this paper is not strong and their experimental performance is weak compared with recently published papers. I do not see a need to have such a new algorithm in the NAS literature. Please see the question below:

Q1. "Bayesnas: a bayesian approach for neural architecture search". ICML 2019
- This paper also takes a pruning's perspective for NAS, but it is much more efficient than the proposed one. Would the authors have some discussion and experimental comparison with this paper? Specifically, Bayesnas considers more complex sparse patterns then the submission.

Q2. "adaptive stochastic natural gradient method for one-shot neural architecture search". ICML 2019
- Could the authors have some discussion with this paper? This paper has comparable performance, but it is also much faster.

Q3. What are the benefits of the proposed method?
- From Tables 2 & 3, the proposed method is not better than STOA on the accuracy or number of parameters.
- CNAS + autoaugmented can offer better accuracy, but the comparison is not fair as different pre-processing method is used.

**Experience Assessment:**

I have published one or two papers in this area.

**Review Assessment: Checking Correctness Of Derivations And Theory:**

I assessed the sensibility of the derivations and theory.

**Review Assessment: Checking Correctness Of Experiments:**

I assessed the sensibility of the experiments.

**Review Assessment: Thoroughness In Paper Reading:**

I read the paper at least twice and used my best judgement in assessing the paper.

---

### Official Review · AnonReviewer4 · 2019-10-30
**Official Blind Review #4**

**Rating:** 3

**Review:**

This paper aims to search a sparse but competitive architecture with using a single fixed type of operation by proposing a channel-level neural architecture search (CNAS). Different from most previous NAS works, this paper conducts NAS process on channel-level such that different cell has different topology. CNAS provides a heuristic algorithm to calculate the saliency vector and zero out the channels iteratively until satisfying a given sparsity. This paper performs CNAS on Cifar-10 and ImageNet, and analyzes the topological properties of the final model. The results of experiment demonstrate CNAS can reach a competitive model with dense models searched by baselines.

This paper provides us with a novel insight that searching neural architecture on the channel level instead of operation and connection level. However, it just combines NAS and pruning parts together, which lacks of novelty in the algorithm level.

I lean to reject this paper because: (1) it lacks of novelty, (2) the experiment result is not convincing, (3) some related works are missed, (4) the expression of the paper is not clear.

Main argument

CNAS is a straightforward combination of NAS and pruning. As the author said in the section 2.1, CNAS method can be seen as two separate processes: training a supernet like one-shot NAS methods and then conducting pruning on the found supernet using Taylor expansion criteria. Both parts are the same as previous works almost and there is no innovation and improvements.

Many related works are missed in the paper. One important step in CNAS is pruning, which uses Taylor expansion technic as previous work. However, it only introduces NAS in introduction section and related works section, ignoring the pruning process. From my view, the pruning part is more important than NAS part.

From the results of the experiment, the improvement of CNAS is not convincing. I think the main focus of the paper is the sparsity, but in Table 2, the number of parameters of model is still larger (4.6*10^6) compared with some baselines like DARTS (2.9*10^6). Besides that, much space in the experiment section is devoted to the relationship between supernet and the final model, which is not so important. Because in other methods, supernet is just an intermedia. Therefore, The comparison between them is not so meaningful.

The paper is hard to understand because of unclear writing. For example, in algorithm part, the author doesn't make DimensionReduction function and its inputs clear. The author mentions the first input in the paragraph but how to combine with the second input? Also, the representation in Figure 1(b) is confused. It's hard to figure out "the thick edges", "the solid thick ones" and "the dotted solid thick ones".

Questions

1. As we all known, operation set is important to the search space. Have you tried more types of operations? From my view, using only one fixed operation is unfair for CNAS compared with other methods.

2. One of your focus is sparsity of the model. Can you explain the reason that you set the number of parameters to a large value (4.6*10^6)? Have you tried to use a larger sparsity value? What's the performance of CNAS when the model is sparser?

3. In Table 2, there are some different tricks (cutout, autoaugmented) applied on some methods. Can you explain how you guarantee the fair comparison between different methods? If we just compare CNAS-R or CNAS-W, they are not better than baselines.


**Experience Assessment:**

I have read many papers in this area.

**Review Assessment: Checking Correctness Of Derivations And Theory:**

I assessed the sensibility of the derivations and theory.

**Review Assessment: Checking Correctness Of Experiments:**

I assessed the sensibility of the experiments.

**Review Assessment: Thoroughness In Paper Reading:**

I read the paper at least twice and used my best judgement in assessing the paper.

---

### Official Review · AnonReviewer3 · 2019-11-04
**Official Blind Review #3**

**Rating:** 3

**Review:**

This paper aims to propose a novel framework for neural architecture search. Although there have been many solutions in the literature, the authors try to build a NAS model that is sparse in structure while being similarly effective as conventional dense models.  The method is straightforward - they select a single fixed operation as edges, and channels as vertices, and the problem of NAS can be directly solved by a gradient descent method. The sparsity can also be achieved on the level of channels.

I have three major concerns, including a lack of novelty, unconvincing experiments, and poor presentation of the work. First, the proposed method is quite straightforward and can be viewed as a quick extension of existing structures. Simplifying the selection of operations make it easy for computation, while it also constrains the applications of the proposed framework. Second, the reported results do not seem to be promising since the improvement was marginal. It is also very difficult to tell whether the contribution is brought by the proposed sparse structure or the adoption of autoaugment since the baseline methods are not applied with it. Third, the paper has not been well written and there are grammatical mistakes throughout the manuscript. I attached the original abstract of the paper and my corrected version below.

There is growing interest in automating designing good neural network architectures. The NAS methods proposed recently have significantly reduced architecture search cost by sharing parameters, but there is still a challenging problem of designing search space. We consider search space is typically defined with its shape and a set of operations and propose a channel-level architecture search (CNAS) method using only a fixed type of operation. The resulting architecture is sparse in terms of channel and has different topology at different cell. The experimental results for CIFAR-10 and ImageNet show that a fine-granular and sparse model searched by CNAS achieves very competitive performance with dense models searched by the existing methods.

There is a growing interest in automating designing good neural network architectures. The NAS methods proposed recently have significantly reduced costs of architecture search by sharing parameters, but there is still a challenging problem of designing search space.
Considering that existing search space is typically defined with its shape and a set of operations, we propose a channel-level architecture search (CNAS) method using only a fixed type of operation.
The resultant architecture is sparse in terms of channels and it has different topologies at different cells.
The experimental results for CIFAR-10 and ImageNet show that a fine-granular and sparse model searched by CNAS achieves very competitive performance with dense models searched by existing methods.


**Experience Assessment:**

I have read many papers in this area.

**Review Assessment: Checking Correctness Of Derivations And Theory:**

I carefully checked the derivations and theory.

**Review Assessment: Checking Correctness Of Experiments:**

I carefully checked the experiments.

**Review Assessment: Thoroughness In Paper Reading:**

I read the paper at least twice and used my best judgement in assessing the paper.

---

### Decision · Program_Chairs · 2019-12-19

**Decision:**

Reject

**Comment:**

This paper proposes a channel pruning approach based one-shot neural architecture search (NAS). As agreed by all reviewers, it has limited novelty, and the method can be viewed as a straightforward combination of NAS and pruning. Experimental results are not convincing. The proposed method is not better than STOA on the accuracy or number of parameters. The setup is not fair, as the proposed method uses autoaugment while the other baselines do not. The authors should also compare with related methods such as Bayesnas, and other pruning techniques. Finally, the paper is poorly written, and many related works are missing.